

# Analysis of small RNA changes in different *Brassica napus* synthetic allopolyploids

Yunxiao Wei, Fei Li, Shujiang Zhang, Shifan Zhang, Hui Zhang and Rifei Sun

Institute of Vegetables and Flowers, Chinese Academy of Agricultural Sciences, Beijing, China

## ABSTRACT

Allopolyploidy is an evolutionary and mechanisticaly intriguing process involving the reconciliation of two or more sets of diverged genomes and regulatory interactions, resulting in new phenotypes. In this study, we explored the small RNA changes of eight F2 synthetic *B. napus* using small RNA sequencing. We found that a part of miRNAs and siRNAs were non-additively expressed in the synthesized *B. napus* allotetraploid. Differentially expressed miRNAs and siRNAs differed among eight F2 individuals, and the differential expression of miR159 and miR172 was consistent with that of flowering time trait. The GO enrichment analysis of differential expression miRNA target genes found that most of them were concentrated in ATP-related pathways, which might be a potential regulatory process contributing to heterosis. In addition, the number of siRNAs present in the offspring was significantly higher than that of the parent, and the number of high parents was significantly higher than the number of low parents. The results have shown that the differential expression of miRNA lays the foundation for explaining the trait separation phenomenon, and the significant increase of siRNA alleviates the shock of the newly synthesized allopolyploidy. It provides a new perspective between small RNA changes and trait separation in the early stages of allopolyploid polyploid formation.

## INTRODUCTION

Polyploidy, or whole-genome duplication (WGD), is prevalent in nature and is particularly common in angiosperms, increasing biodiversity and providing new genetic material for evolution (*Wendel, 2000*). Synthetic polyploidy is often associated with novel and presumably advantageous ecological attributes such as range expansion (*Hijmans et al., 2007*), novel secondary chemistry and morphology (*Leitch & Leitch, 2008*), and increased pathogen resistance (*Nuismer & Thompson, 2001*). Previous studies have investigated synthetic allopolyploids and show that various genetic (*Song et al., 1995*; *Xiong, Gaeta & Pires, 2011*) and epigenetic (*Adams et al., 2003*; *Cui et al., 2013*; *Ge, Ding & Li, 2013*) changes, as well as alterations in gene expression levels (*Wang et al., 2006*; *Chelaifa, Monnier & Ainouche, 2010*; *Yoo, Szadkowski & Wendel, 2013*) occur at the initial stage of allopolyploidization. At the genetic level, loss of parental and/or appearance of novel

Corresponding author
Rifei Sun,
yuluoyunxiao@outlook.com

sequences at the initial stage of allopolyploidization are common events. Non-homologous chromosome exchanges occur in synthetic *B. napus*, resulting in the addition and/or deletion of sequences (*Gaeta et al., 2007*). At the epigenetic level, changes in small RNA and DNA methylation patterns occur at the initial stage of allopolyploidization. Shen et al. reported higher siRNA and DNA methylation levels in F1 hybrids (*Shen et al., 2017*). The role of heredity and epigenetics leads to changes in gene expression, which in turn leads to novel phenotypes (*Chen, 2007*).

Non-coding small RNAs are widely found in eukaryotes, which are endogenous with a length of about 20–24 nt. Many studies have shown that small RNAs play an important role in gene expression regulation through transcriptional level gene silencing, or post-transcriptional level gene silencing (*Baumberger & Baulcombe, 2005*). Their first report was the phenomenon of RNA interference in nematodes (*Lee, Feinbaum & Ambros, 1993*), and later the phenomenon of gene silencing or inhibition was discovered (*Napoli, Lemieux & Jorgensen, 1990*; *Carvalho et al., 1992*; *Hannon, 2002*). Shortly after these studies, the researchers confirmed that post-transcriptional gene silencing in plants is associated with small RNA activity (*Hamilton & Baulcombe, 1999*). These small RNAs regulate various biological processes by interfering with the translation of mRNA. In plants, small RNAs can be divided into two major categories depending on their synthesis and function: miRNA and siRNA. miRNAs and siRNAs are considered to be highly conserved and are important gene expression regulators in plants (*Jones-Rhoades, Bartel & Bartel, 2006*; *Axtell & Bowman, 2008*). Small RNA is a molecule and the approach is a molecular biological tool to control gene expression at the transcriptional and post-transcriptional levels.

Oilseed rape (*B. napus*, AACC, $2n = 38$), which is generally thought to be naturally crossed and doubled between *B. rapa* (AA, $2n = 20$) and *B. oleracea* (CC, $2n = 18$), was formed 7500 years ago and is a good model for exploring allopolyploids (*Nagaharu & Nagaharu, 1935*; *Chalhoub et al., 2014*). In addition, oilseed rape is now one of the most important oilseed crops in the world and is inseparable from people's lives. However, due to the short history of domestication between 300 and 400 years ago, the genetic basis of oilseed rape was narrower than that of the parental species (*Go'mez-Campo, 1999*), which further led to the restriction of oilseed rape breeding and utilization of heterosis. Therefore, it is necessary to explore the molecular mechanism of the distant hybridization between *B. rapa* and *B. oleracea* in order to obtain synthesis *B. napus* and to expand the germplasm resources of *B. napus*.

Previous studies have explored sRNA changes and regulatory patterns in different generations of resynthesized *B. napus* (*Fu et al., 2016b*). However, these patterns in allopolyploids with different traits have not been fully examined to answer the relationship between trait differences and small RNA changes. The present study analyzed small RNA changes of eight F2 synthetic *B. napus*. We found that a part of miRNAs and siRNAs were non-additively expressed in the synthesized *B. napus* allotetraploid and the number of siRNAs present in the offspring was significantly higher than that of the parent, which can alleviate the effects of "genome shock". Differentially expressed miRNAs and siRNAs differed among eight F2 individuals. The differential expression of miR159 and miR172 was consistent with that of flowering time trait.Differential expression of small RNAs further

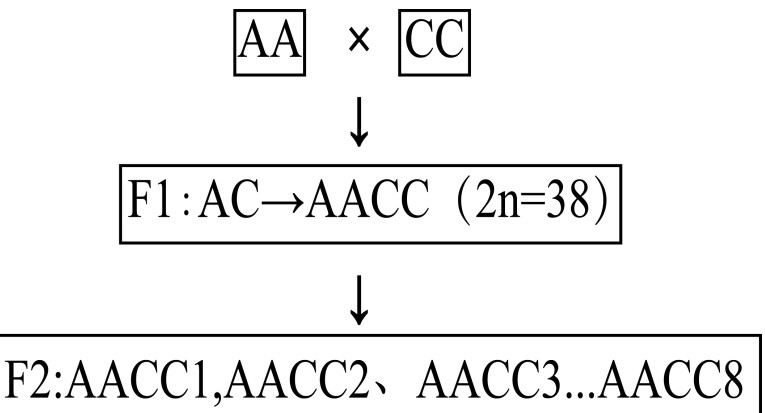

**Figure 1  Plant materials.**

affects the expression of the traits by affecting the expression of the target genes, thereby explaining the phenomenon of partial trait separation. It provides a new perspective of small RNA changes and trait separation in the early stages of allopolyploid formation.

## MATERIAL AND METHODS

### Plant materials

For this study, we used 10 accessions, including the female parent Cai-Xin, male parent Chinese kale, and eight F2 synthetic allopolyploids (Fig. 1). First, by embryo rescuing, F1 haploid (AC) hybridization between Cai-Xin (P1) and Chinese kale (P2) was performed. Then, F1 allopolyploids (AACC) were obtained by colchicine doubling (*Wei et al., 2017*). Seeds were collected by F1 (AACC) budding self-pollination. The eight F2 plants and the parents were planted in the greenhouse of the Chinese Academy of Agricultural Sciences Institute of Vegetables and Flowers (Beijing, China). We investigated the field traits during the flowering period: flower time and flower size.

### sRNA library construction and sequencing

Young leaves next to bud (five cm in length) were collected, frozen in liquid nitrogen, and stored at −80 °C until extraction. RNA was extracted from three biological replicates using TRIzol reagent (Invitrogen, Life Technologies) following standard protocols. The quality and quantity of the extracted RNA were assessed using the agarose gel electrophoresis, NanoPhotometer® spectrophotometer, Qubit and Agilent 2100 (Beijing China). Then a linker was added to both ends of the small RNA and reverse transcription to synthesize cDNA. Subsequently, after PCR amplification, the target DNA fragment was separated by PAGE gel electrophoresis, and the cDNA library was recovered by gelatinization. The final PCR products were sequenced using Hiseq 2500 at Nuohe company (Beijing, China).

### Identification of miRNAs and siRNA clusters

After trimming adaptor sequence at the 5′ and 3′ ends of the sequenced reads, the cellular structural RNAs (e.g., rRNAs, snoRNAs, snRNAs) were removed using in-house Perl

scripts. Clean reads of 18 to 30 nt were aligned to the *B. rapa* (*Wang et al., 2011*) and *B. olearea* (*Liu et al., 2014*) genome with the bowtie2 software (*Langmead & Salzberg, 2012*) with the parameter setting for a perfect match. The sequences of P1 were aligned to the *B. rapa* (*Wang et al., 2011*), the sequenses of P2 were aligned to *B. oleracea* (*Liu et al., 2014*), and the sequences of F2 were aligned to the merge genome of *B. rapa* and *B. olearea* (*Wang et al., 2011*; *Liu et al., 2014*). A miRNA was considered as conserved if its mature sequence had two or fewer nucleotide mismatches compared with the miRNAs in miRBase (http://www.mirbase.org, release 21) (*Meyers et al., 2008*).

After removing sRNAs aligned tomiRNA, the remaining sRNA reads were then used to identify siRNA clusters. Only reads mapped to unique loci were counted for subsequent analyses. A siRNA cluster or locus was defined as a genomic region matched by at least three sRNA reads. If one cluster resided within 200 nt of another, they were merged and regarded as a single cluster.

## Differential expression analysis of miRNA and siRNA

sRNAs were counted as miRNA reads when they were fully or partly ($\geq 1$ nt) overlapping with the mature miRNA sequence. The expression levels of miRNAs were normalized to reads per million (RPM) that was calculated using the formula RPM = number of miRNA reads /total number of clean reads $\times 10^6$.

The expression level of a siRNA cluster was estimated by uniquely mapped reads. Only reads with a full-length perfect match were accepted as hits. And the expression levels were normalized to reads per million (RPM) for further analysis. The expression differences of miRNAs and siRNA clusters were determined by DEGseq (*Wang et al., 2010*).

## miRNA target prediction and GO enrichment analysis of target genes

The targets of miRNAs in *B. rapa* and *B. olearea* were predicted using psRNATarget (http://plantgrn.noble.org/psRNATarget/; *Dai, Zhuang & Zhao, 2018*). Default parameters were used to filter candidates.

GO enrichment analysis was implemented by the GOseq R package (http://www.bioconductor.org/packages/release/bioc/html/goseq.html) in which gene length bias was corrected. AgriGO (a Web-based tool and database for gene ontology analysis; http://bioinfo.cau.edu.cn/ agriGO/) (*Du et al., 2010*), was also used in this study. GO terms with a corrected FDR $\leq 0.05$ were considered to be significantly enriched.

## Data accessibility statement

The small RNA data we sequenced would be uploaded to genebank database after the article is published.

## RESULTS

### sRNAs of resynthesized *B. napus* and its parents

The reads obtained by parental sequencing were 16.1 million and 14.0 million, respectively. The reads of eight F2 plants were 15.3 million, 15.2 million, 15.8 million, 13.2 million, 16.0 million, 18.3 million, 14.5 million and 11.0 million respectively (Fig. 2A). After
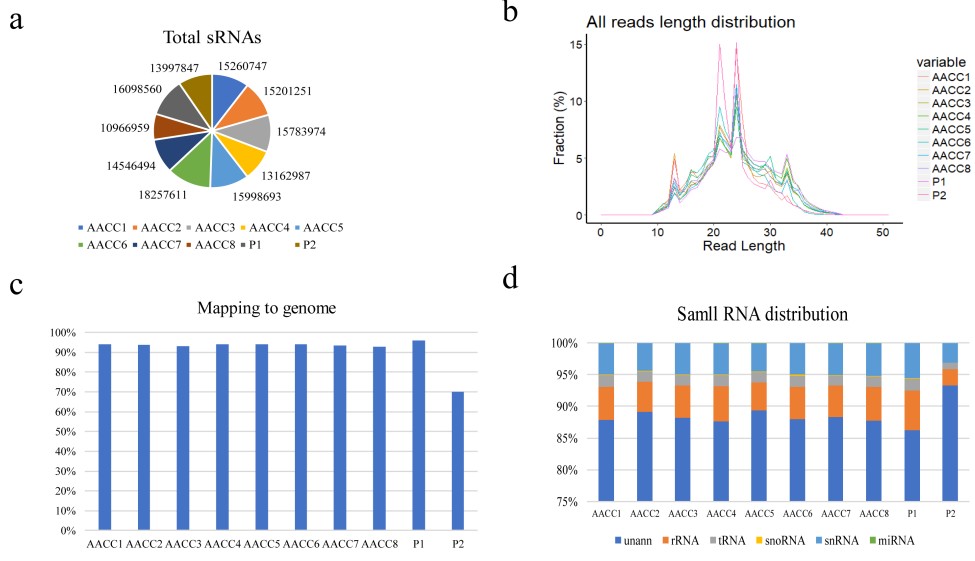

**Figure 2  Small RNA sequencing-quality analysis and length distribution.** (A) Number of clean reads for small RNA sequencing in each sample; (B) distribution of small RNA in each sample; (C) ratio of all sRNAs in each sample to the genome (*B. rapa* and *B. olearea*); (D) annotated information of small RNA in each sample and its proportion to total small RNA.

removing low quality reads, 10.6–15.7 million clean reads were obtained (96.79%–98.97% of total reads). Among them, 21–24 nt reads account for the largest proportion, which was 15%–16% in the ten samples (Fig. 2B). The reads size distribution is not the same with that of previous reports (*Fu et al., 2016b*), which may be due to different materials used in the experiment.

Average 93.69% of the total reads in the eight F2 samples, 96% and 70% of two parents samples, could be perfectly mapped to the *B. rapa* genome and *B. olearea* genome with no mismatch (Fig. 2C). The alignment ratio of the eight F2 samples was relatively high, which was inconsistent with previous research results (*Shen et al., 2017*). May be because the method of comparison was different. *Shen et al. (2017)*'s experimental method was to compare the sequencing data with the genome of *B. napus*. This experiment was to compare the sequencing data with the genome of *B. rapa* and *B. oleracea*. The high ratio indicates that our method was reasonable and the utilization of reads was relatively high. In addition, the perfectly mapped reads consisted of various types of sRNAs, including miRNA, rRNAs, tRNAs, snoRNAs, snRNAs, and unannotated sRNAs (Fig. 2D). A large fraction of perfectly mapped unique reads (approximately 88.58%) was not annotated and probably includes new siRNA candidates (Fig. 2D).

## Identification and comparison of miRNAs in eight synthetic *B. napus* and its parents

We identified known miRNAs by aligning clean reads with miRBase databases 22.1. 72050–173239 known miRNAs of eight F2 samples, 22208 and 1410 known miRNAs of two parent samples were detected. The number of miRNAs found in F2 was much higher

than that of the parent, indicating that hybridization and polyploidization caused changes in the number of miRNAs (Table S1). Due to the impact of the "genome shock", the miRNAs of the hybrid progeny vary greatly, and the miRNA relate to gene regulation during the physiological process plays a role in relieving the impact of the "genome shock" (*Guan et al., 2014*; *Ghani et al., 2014*; *Shen et al., 2015*).

To further explore the changes in miRNAs caused by hybridization and doubling, we compared the expression quantity between eight F2 plants and mid-parent value (MPV). Under the assumption of additive expression, average expression in each of the eight F2 samples was quite different with the MPV.

A total of 33 conserved bra-miRNAs in the eight F2 samples were found to be non-additively expressed ($P \leq 0.05$, FDR $\leq 0.05$). Of the 33 miRNAs, 11, 1, 9, 8, 5, 4, 11 and 6 were non-additively activated in eight F2 allotetraploids, respectively, while 7, 8, 5, 7, 7, 8, 6 and 5 were non-additively repressed in eight F2 allotetraploids, respectively (Fig. 3A). A total of 13 conserved bol-miRNAs in the eight F2 samples were found to be non-additvely expressed ($P \leq 0.05$, FDR $\leq 0.05$). Of the 13 miRNAs, 5, 5, 2, 3, 3, 4, 4 and 4 were non-additively activated in eight F2 allotetraploids, respectively, while 5, 4, 7, 5, 5, 6, 8 and 5 were non-additively repressed in eight F2 allotetraploids, respectively (Fig. 3B). The target genes of these non-additively expressed miRNAs were predicted from the gene models in the *B. rapa* and *B. olearea* genome annotation via psRNATarget analysis (*Wu et al., 2009*). A total of 300 target transcripts were predicted for 33 conserved bra-miRNAs. A total of 269 target transcripts were predicted for 13 conserved bol-miRNAs.

Compared with all annotated *B. rapa* genes, these target genes were significantly ($P < 0.01$) enriched for 16 biological process, 4 cellular component and 19 molecular function GO terms, including the ATP biosynthetic process, phospholipid transport and auxin efflux transmembrane transporter activity (Fig. 3C). Compared with all annotated *B. olearea* genes, these target genes were significantly ($P < 0.01$) enriched for 6 biological process, 1 cell cellular component and 8 molecular function GO terms, including phospholipid transport, ATP biosynthetic process and ATPase activity, coupled to transmembrane movement of ions, phosphorylative mechanism (Fig. 3D).

These results suggest that the protein amino acid phosphorylation pathways and miRNA-mediated regulation of genes of the pathways might be a potential regulatory process contributing to heterosis.

## Identification and comparison of siRNAs in eight synthetic *B. napus* and its parents

To characterize the effects of siRNAs on genome stability and gene expression, we surveyed siRNA density and expression level in different allotetraploids. In eight F2 samples, about 18744–38962 siRNA clusters were identified. 19 and 101 siRNA clusters were identified in two parents. To further investigate the expression level of the siRNA clusters between F2 and the two parents, we found that high_parent in the eight F2 samples were 38963, 26014, 27654, 20226, 22592, 37995, 25629, 18746, much higher than low_parent 81, 81, 80, 80, 80, 80, 79, 81 (Fig. 4C). Most of the siRNAs in the progeny showed high_parent expression. The results indicate that hybridization and polyploidization caused a surge in
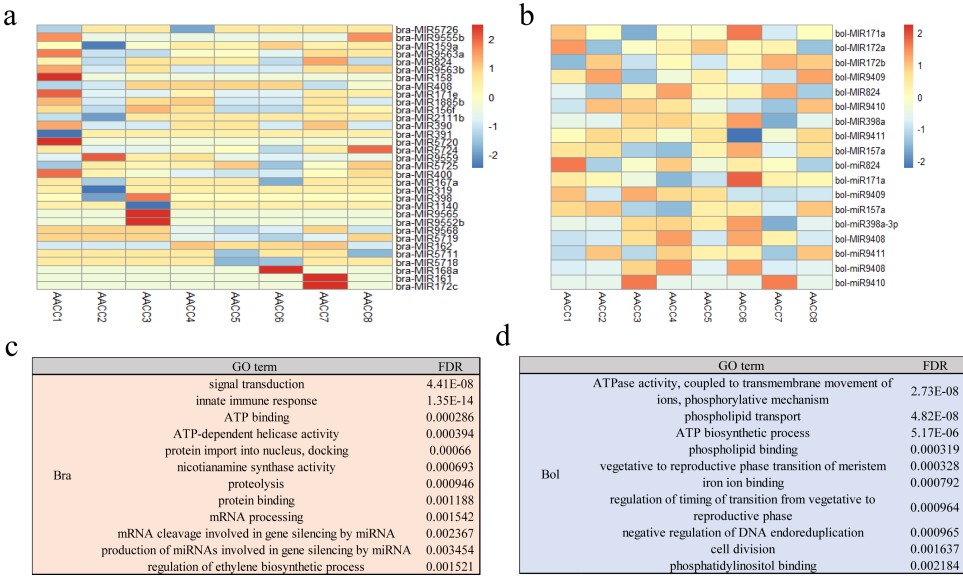

**Figure 3** **miRNAs and their targets in the F2 hybrids.** (A) Expression changes of the differentially expressed conserved miRNAs in the F2 hybrids compared with the MPV when aligning *B. rapa* genome; (B) Expression changes of the differentially expressed conserved miRNAs in the F2 hybrid compared with the MPV when aligning *B. olearea* genome; (C) Gene Ontology terms of the differentially expressed miRNAs targets in F2 hybrids when aligning *B. rapa* genome; (D) Gene Ontology terms of the differentially expressed miRNAs targets in F2 hybrids when aligning *B. rapa* genome.

siRNA. siRNA has the effect of mitigating polyploidy shock, which was consistent with the previous conclusions (*Fu et al., 2016b*).

In addition, the number of siRNAs in eight F2 plants was 249275 –516295, which was much higher than the number of parents (1260) (Fig. 4A). AACC1 had the highest number of siRNAs, followed by AACC6. The results showed that there was a difference in number of siRNA clusters in eight F2 plants.

Based on comparison of expression levels, we found that 387–679 of the siRNA clusters were differentially expressed between F2 and its two parents ($P \leq 0.05$, pairwise Student's $t$-test). Interestingly, to AA genome, the number of siRNA clusters in F2 plants with an expression level higher than the MPV (220, 194, 172, 221, 225, 307, 149, 174 in eight F2 samples respectively) was significantly higher than the number of siRNA clusters with an expression level lower than the MPV (8, 7, 7, 3, 3, 7, 7, 7 in eight F2 samples respectively). To CC genome, the number of siRNA clusters in F2 plants with an expression level higher than the MPV (208, 130, 151, 177, 162, 242, 126, 113 in eight F2 samples respectively) was significantly higher than the number of siRNA clusters with an expression level lower than the MPV (76, 75, 77, 51, 50, 77, 77, 76 in eight F2 samples respectively) (Fig. 4B). In both *B. rapa* and *B. olearea* genomes, the number of up-regulated siRNAs expression in F2 plants is much higher than the number of down-regulated expression. The results showed that there are more up-regulated expression siRNAs to AA genome, while there are more down-regulated expression siRNAs to CC genome. Therefore, two genomes (AA\CC genomes) showed significant differences in response to WGD. It is indicated that

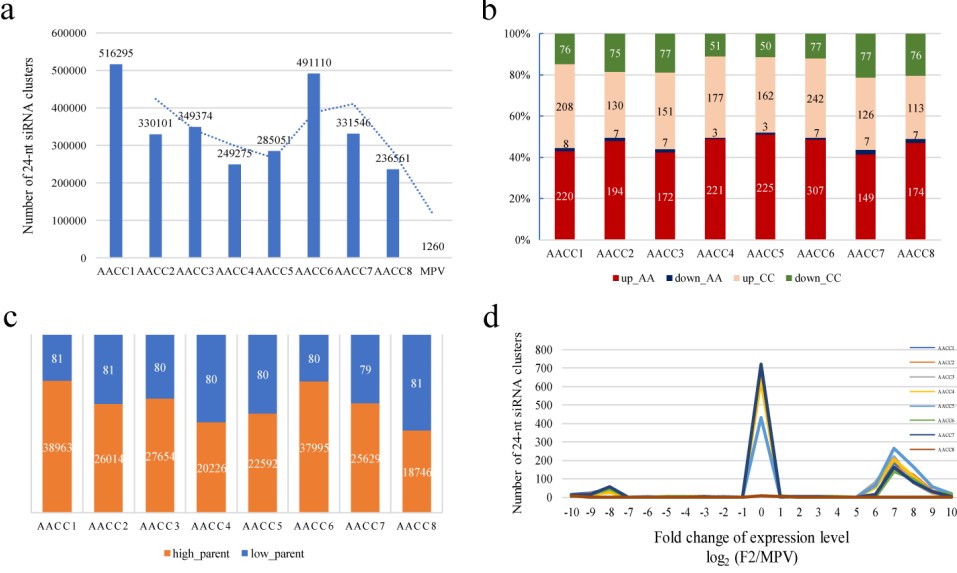

**Figure 4** **Expression patterns of the siRNA clusters in F2 hybrids.** (A) Number of the siRNA clusters in F2 hybrids and MPV; (B) Number of siRNA clusters that were up- or down-regulated in F2 hybrids compared with MPV; (C) Number of siRNA clusters that were high- or low-parent in F2 hybrids; (D) Distribution of the number of the siRNA clusters based on their expression level changes between F2 hybrids and MPV.

hybridization and polyploidization produce many new siRNAs, which can alleviate the effects of "genome shock".

Consistent with this observation, the majority of siRNA clusters had a log2(F1/MPV) value of 7 (Fig. 4D), significantly deviating from the null expectation [log2(F1/MPV) = 0]. This result indicates that the distribution trend of eight F2 plants was consistent, AACC1 changed the most, and AACC8 changed the least. In short, there was a difference in siRNA differential expression of eight F2 plants. Because the newly formed allotetraploid is unstable, there are differences in the small RNA variation of different F2 plants, which lays a foundation for the performance of different plant traits.

## Flowering trait analysis

Among the differentially expressed miRNAs, we found that bra-miR159 was down-regulated in AACC2 and up-regulated in other individuals. We used psRNATarget to predict the 14 target genes of bra-miR159 online, which were MYB101 and MYB65 transcription factors. These two transcription factors promote the expression of the LHY gene, which in turn promotes flowering (*Liu & Chen, 2009*; *Wu et al., 2009*; *Anwesha & Thomas, 2010*) (Fig. 5A). In addition, bol-miR172 was up-regulated in AACC1 and down-regulated in other individuals. It was predicted that bol-miR172a has 6 target genes, which were AP2-like and AP2 transcription factors. These two transcription factors inhibit the expression of FLOWERING LOCUS T (FT) (*Wu et al., 2009*; *Wollmann et al., 2010*; *Zhu & Helliwell, 2011*), which in turn delays flowering (Fig. 5B). The results of field traits

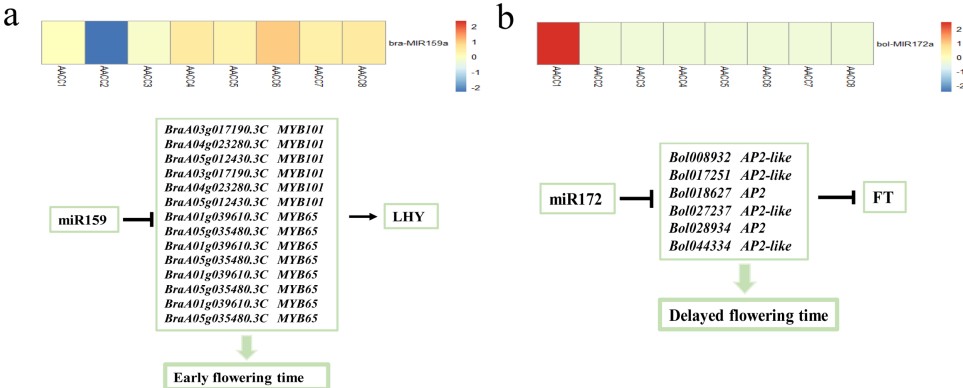

**Figure 5  Analysis of miR159 and miR172 in F2 hybrids.** (A) miR159 difference analysis in the F2 hybrids; (B) miR172 difference analysis in the F2 hybrids.

showed that AACC1 flowered at the earliest and AACC2 flowered earlier. The results show that small RNA could affect the trait by regulating the expression of the target gene.

## DISCUSSION

### Changes in miRNA in the eight synthetic *B. napus* allopolyploids

Small RNA plays an important role in polyploid inheritance and gene expression by altering chromatin structure and regulating gene expression. Therefore, studying the expression level of small RNA is helpful to study the regulation mechanism of polyploid gene expression (*Yao et al., 2007*; *Ng, Lu & Chen, 2012*; *Xie & Zhang, 2015*). Studies have analyzed the expression and distribution of miRNAs and siRNAs in allotetraploids of *A. thaliana* and found that small RNAs act as a buffer to buffer the genomic shocks of Arabidopsis polyploids (*Ha et al., 2009*). Xie et al. studied *G. hirsutum* (genomic AADD) and found that cotton has an increased miRNA relative to its two diploid ancestors (*Xie & Zhang, 2015*). In addition, in the Brassica, Fu et al. found that the number and expression levels of miRNAs in the newly synthesized *B. napus* from different generations increased compared with the parents (*Fu et al., 2016a*). These conclusions are consistent with our findings, indicating that the process of hybridization and doubling of the multiploidization results in a general increase in miRNAs. In addition, we found that the number of miRNAs in the newly synthesized *B. napus* was much higher than that of the parents. Loss of miRNAs and new phenomena also exist (Table S1 ). Our results are inconsistent with the results of *Fu et al. (2016a)*'s research, possibly due to material differences: the parent we use was the follower Chinese cabbage, and the parent used by *Fu et al. (2016a)* was *B. campestris*. It is speculated that due to genetic differences among different subspecies, the number of differentially expressed miRNAs in newly synthesized *B. napus* is different.

Besides, there are few reports on the changes in sRNAs between different traits in allopolyploid. It was found that the number of miRNAs differed between different individuals. Some miRNAs are expressed much higher in one F2 plant than orther F2 plants. Some miRNAs are expressed in one F2 plant and are not expressed in another

F2 plant. *Zhang et al. (2016)* analyzed the transcriptomes of different individual plants of synthesis *B. napus*, and found that there were differences in gene differential expression. This experiment shows that miRNAs also differ between different plants. This result indicates that the initial genome of allopolyploid formation was unstable, and there are differences in miRNA expression in self-crossing progeny.

All of these results suggest that miRNAs play important roles in the regulation of interspecific hybridisation and polyploidization processes, and hence in subsequent evolution of polyploidy crops (*Fu et al., 2016a*; *Fu et al., 2016b*; *Shen et al., 2015*; *Shen et al., 2017*).

## Insights into non-additively miRNA regulation in the synthetic *B. napus* allopolyploids

Non-additively gene expression occurs during polyploidization, which results in the complexity of gene expression and phenotye. The miRNA regulate the expression of target molecules (mRNA). Therefore, the miRNA gene is non-additively expressed or is itself mutagenized to form a base mutation, which may lead to the generation of a new target site or the loss of the old target site, resulting in a new phenotype. The comparative analysis of miRNAs and siRNAs between *A. arenosa* and their parents showed that the expression patterns were highly variable between tetraploids and diploids, and most miRNAs were non-additively (*Ha et al., 2009*). *Li et al. (2014)* simulated the early generations of common wheat by using new synthetic wheat derived from crossing between *Triticum turgidum* (AABB) and *Agilops tauschii* (DD) and chromosome doubling. It was found that a high proportion of miRNAs were non-additively expression in synthetic wheat, which resulted in differential expression of important target genes. Dynamic regulation of some homologous genes mediated by small RNA may be responsible for the heterosis of new hexaploid wheat (*Li et al., 2014*). In our study, high-throughput sequencing was used to compare miRNA expression between eight F2 plants and their parents. Our results show that 48.9% of miRNAs are non-additively expressed in the synthetic *B. napus*. (*Fu et al., 2016a*; *Fu et al., 2016b*) found that approximately 86.6% of miRNAs were non-additively expressed in different generations of synthetic *B. napus* (*Fu et al., 2016a*; *Fu et al., 2016b*). The reason for this difference may be the different materials used in the experiment. Fu's experimental materials were different plants of different generations, and our materials were different plants of the same generation. The experimental results show that the differences between generations may be greater than the differences between the same generation.

In addition, small RNA affects the character performance by regulating the expression of target genes. Guan et al. found miR828 and miR858 regulate homoeologous MYB2 gene in polyploidy cotton, which in turn affects the fabric trait (*Guan et al., 2014*). In allopolyploid Arabidopsis, target genes of miR163 encode a family of small molecule methyltransferases involved in secondary metabolite biosynthetic pathways (*Ng et al., 2011*). In our experiment, non-additively expression differs among eight plants. The differential expression of bra-miR159 and bol-miR172 in eight F2 plants was consistent with the results of flowering time in the field, which further explained the differences in flowering traits among eight individuals. The transcription factors of bra-miR159

are MYB101 and MYB65, which in turn promote the expression of the LHY gene. The transcription factors of bol-miR172 are AP2-like and AP2, which in turn inhibit the expression of FLOWERING LOCUS T (FT). Eventually, changes in small RNA lead to changes in flowering time. The results further indicate that epigenetic variation is widespread in the early stage of allopolyploid formation. Differential expression of small RNAs further affects the expression of the traits by affecting the expression of the target genes, thereby explaining the phenomenon of partial trait separation.

The non-additively expression of miRNAs in synthetic heteropolyploids may increase polyploid fitness. In Arabidopsis allotetraploids, non-additively expressed genes are involved in multiple biological processes, which may provide an evolutionary mechanism for heterologous polyploid selection and adaptability (*Wang et al., 2006*). In our experiments, the relevant target genes of non-additively expressed miRNAs were significantly enriched in ATP-related pathways. These findings suggest that non-additively mRNA/miRNA may play an important role in the growth and non-additively phenotypes of polyploids.

### Changes in siRNA in the eight synthetic *B. napus* allopolyploids

siRNA-mediated epigenetic mechanisms have been shown to be important for maintaining the genomic stability of allopolyploids (*Misook et al., 2009*; *Lu et al., 2012*), and the increasing of siRNAs has the effect of slowing geome shock. siRNA is generally produced by endogenous transposons and repetitive sequences in plants. In addition, siRNA is non-additively expressed in newly synthesized allopolyploids, which may promote the variation and adaptability of polyploids (*Shen et al., 2017*). In this study, the relationship of the siRNA clusters expression levels between eight F2 plants and parents were studied. The number of up-regulated expression of siRNA cluster between eight F2 plants and the parents was found to be greater than the number of down-regulated expression. *Ha et al. (2009)* found that the newly synthesized Arabidopsis allopolyploid siRNA clusters were reduced. This indicates that the expression patterns of siRNA among different species are not the same, and the expression of siRNA of other allopolyploids needs further investigation. Another possibility is speculated to be related to different experimental analytical methods. Previous research generally use allopolyploid genomes (rapeseed, Arabidopsis, and cotton) for data aligning. The experimental method is to align with the parents's (*B. rapa* and *B. olearea*) genomes respectively. In the early stage of allopolyploid formation, genomic information of synthetic allopolyploid was closer to the parents' genome than to the allopolyploid genome after evolution, so the method could identify more differences. In addition, the number of non-additively expressed siRNAs in eight F2 plants was different, AACC1 had the largest variation and AACC8 had the smallest variation. The experimental results further indicated that the siRNA increased significantly in the early stage of allopolyploid formation to alleviate the genome shock, and there was a difference between the different allopolyploids, which may be related to the trait separation.

## CONCLUSIONS

The study explored the small RNA changes of eight F2 synthetic *B. napus* using small RNA sequencing. The results have shown that the differential expression of miRNA lays the foundation for explaining the trait separation phenomenon, and the significant increase of siRNA alleviates the shock of the newly synthesized allopolyploidy. It provides a new perspective between small RNA changes and trait separation in the early stages of allopolyploid polyploid formation.

## ACKNOWLEDGEMENTS

The experiment was performed at the Key Laboratory of Biology and Genetic Improvement of Horticultural Crops, Ministry of Agriculture, Beijing, China. We would like to thank Accdon for providing linguistic assistance during the preparation of this manuscript.

### Funding

This research was funded by the National Key Research and Development Program of China (grant number 2017YFD0101802) and the Fundamental Research Funds for Central Non-profit Scientific Institution (grant number IVF-BRF2018003). The funders had no role in study design, data collection and analysis, decision to publish, or preparation of the manuscript.

### Grant Disclosures

The following grant information was disclosed by the authors:
National Key Research and Development Program of China: 2017YFD0101802.
Fundamental Research Funds for Central Non-profit Scientific Institution: IVF-BRF2018003.

### Competing Interests

The authors declare there are no competing interests.

### Author Contributions

- Yunxiao Wei performed the experiments, analyzed the data, prepared figures and/or tables.
- Fei Li, Shujiang Zhang, Shifan Zhang and Hui Zhang contributed reagents/materials/-analysis tools.
- Rifei Sun conceived and designed the experiments.

### Data Availability

The raw measurements are available in the Supplemental Information. The small RNA data is available at SRA: SAMN12569366, SAMN12569367, SAMN12569368, SAMN12569369, SAMN12569370, SAMN12569371, SAMN12569372, SAMN12569373, SAMN12569374, SAMN12569375.

## Supplemental Information

Supplemental information for this article can be found online at http://dx.doi.org/10.7717/peerj.7621#supplemental-information.

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
