# Peer review of "Analysis of small RNA changes in different Brassica napus synthetic allopolyploids"

_PeerJ, doi:10.7717/peerj.7621_

## Round 0.1 · original submission · Major Revisions

I feel the article is still need of revision and I would be happy to accept the manuscript if the authors have carried out the necessary revision.

·

Basic reporting

The manuscript needs more careful editing. It is very poorly prepared. Literature references, significant background of the work are provided. Article structure, figures, tables and additional data are fine. However, reference list is very poorly prepared. Uniformity in format, complete details such as page numbers are missing. The manuscript is self contained with relevant results to hypotheses tested.

Experimental design

The research is original and fits with the aims and scope of the journal. Research question is defined well and meaningful. Role of small RNAs in polyploidization is less studied and hence the work fills a wll identified knowledge gap. Variations among the F2 lines analyzed are interesting. However, needs better discussion. The techical part of the work such as experments and ethical standards are high. Methods described are sufficient for replication. The field trails for observing the flowering time should be elaborated in materials and methods and discussed in detail in discussion section.

Validity of the findings

Some of the conclusions made are blunt statements. Authors have to discuss in detail about the molecular mechanism of flowering, role of small RNAs and the target transcription factors and other genes regulated by transcription factors.

Additional comments

The manuscript entitled 'Analysis of small RNA changes in different Brassica napus synthetic allopolyploids' is a good work overall but suffers from some serous issues and some minor issues detailed below. The manuscript needs careful editing and also elaborated discussion.

Major points:
1. The procedure followed to separate and confirm siRNA from other sRNA in the sequence reads. Is it the usual procedure? Any reference?
2. There are many blunt statements in results and discussion. It has to be elaborated with literature support.
Eg. Line 148. What is meant by method of comparision?
3. Lines 157-158. Polyploidization and hybridization caused changes in number of miRNA. Why it should change? Relate to gene regulation during the physiological process with reference.
4. Line 180. How could protein phosphorylation contribute to heterosis? Elaborate with cited references on what is happening to proteins during heterosis and to which proteins.
5. Line 190. What did Fu etal 2016a said? How you state it is consistent with that? Elaborate.
6. Lines 201-204. Number of up-regulated siRNA expression in F2 compared to number of down-regulated siRNA expression indicate what? Why should there be a difference in differential expression of siRNAs amoung F2 plants? This could have been discussed better.
7. Lines 211-221. Observation of flowering time in F2 plants in field trial is not enough to support the argument about role of transcription factors and their targeting miRNAs. A simple RT-PCR to confirm the suppression of AP2-like and AP2 transcription factors would have given a strong support.
8. Line 241. Inconsistent and consistent are not appropriate words. It can be used only when same samples are repeated and when we get different results compared to previous reports. It is obvious that when the subspecies used are different, the results are going to be different from previous reports. There is no surprise. There is no point in arguing consistent or inconsistent.
9. Lines 242-248. This should have been discussed in a much better way. Gene expression and regulation during initial stages of allopolyploid formation - needs referenes to support detailed arguments.
10. Line 252. miRNAs do not regulate sequence of target site. They complement and thus regulate the expression of target molecules (mRNA).
11. Lines 253-255. Unclear. miRNA gene undergoes mutation thereby the sequence changes and loses its complementarity to target mRNA. This can have an impact on the phenotype indirectly or may involve in evolution of new phenotype. Due to mutation in miRNA, its target site (mRNA) may not change or get lost.
12. Lines 265-268. Why there should be difference between different materials of same generation and different materials of different generation? Give a scientific argument or at least a speculation.
13. Lines 271-282. A given miRNA may have many target mRNAs. Since one of the target is a transcription factor involved in regulating flowering time (as per literature / database), the flowering observed cant be just correlated with teh miRNA-mRNA combination. It needs validation. Include discussion on what are the genes regulated by this transcription factor and how many of them are involved in control of flowering time.
14. References are poorly prepared. Journal names are not given in full. Abbreviated journal names are not used in many references. Volume number or page numbers missing in most of the references. There are many typographical errors in reference list.

Other minor points:
1. Line 65-66. Avoid use of 'means of molecular biology'. Small RNA is a molecule and the approach is a molecular biological tool. Change the words appropriately.
2. Line 74. 'Stable synthesis B. napus'?
3. Line 77. Change 'are not been' to 'have not been'
4. Line 83. Remove the word 'polyploid'
5. Lines 88-89. Embryo rescue and in vitro hybridization. How this was performed. Elaborate the methodology so that it can be replicated by other researchers.
6. Line 93. Change to 'and flower size'
7. Line 99. Quit and Agilent 2100 should be within brackets? Include city and country name as well.
8. Line 110. Check spelling of the word 'sequences'.
9. Line 114. Change to 'miRNA'.
10. Line 114. Instead of 'we got', use the words 'were obtained' at the end of the sentence.
11. Line 144. What is meant by different materials? plants or tissues?
12. Line 145. 'on average, average'
13. Line 155. Change to 'identified'
14. Line 156. Change to 'known'
15. Lines 162-165. What is bra-miRNA and bol-miRNA?
16. Line 173. 'cell cellular component'??
17. Lines 163-171. 'Non-additively activated'. Explain.
18. Line 188. Change to 'indicate'
19. Line 192. Change to 'followed by AACC6'
20. Line 192. 'number of parents' or 'number in parents'?
21. Lines 238, 239. Change to Fu et al. (2016a).
22. Line 255. A. thaliana... the statement has to be rephrased.
23. Line 257. Add year to Li et al.
24. Line 259. 'high propotion of miRNAs were non-additively expression' - poor English. Where found to be non-additively expressed?
25. Lines 260-261. Which genes? Examples?
26. Line 263. Add year to Fu et al.
27. Line 264. What is meant by non-expressively expressed??
28. Line 269. 'performance of field traits'??
29. Lines 273, 277, 278. 'non-additive expression' 'non-additively expressed'. Check.
30. Line 274. There are hundreds of field traits. What is again meant by consistent?
31. Line 297. 'Predecessors generally use..."??
32. Lines 283-300. Discuss in a better way.
33. Line 302. Check the spelling of author (auther). Contribution should have lower case c.
34. Line 304. 'contributed to the' is repeated appearing two times.
35. Line 318. Data Accessiblity Statement. Use lower case A and S for accessibility and statement.
36. Line 322. Journal name should be 'Proc Natl Acad Sci USA'
37. Line 325. Page numbers or DOI is missing for this reference.
38. Line 327. 'Trands' should be 'Trends'. Page numbers missing. Period after the article title is missing.
39. Line 329. Period after the article title is missing. 'U S A' should be 'USA'. Page numbers missing.
40. Line 330. Page numbers missing.
41. Line 344. Nucleic acids research should be 'Nucleic Acids Res'. Volume and page numbers missing.
42. Line 346. Give abbreviated journal name.
43. Line 355. Check the format in instructions to authors or recent issue of Peer J. Page numbers missing.
44. Line 359. Journal name is given in full. USA is missing. Volume and page numbers missing.
45. Line 362. Page numbers missing.
46. Line 364. Page numbers missing.
47. Line 368. Page numbers missing.
48. Line 371. Page numbers missing.
49. Lines 371 and 374. In all other references, title of the papers are given in lower case. In these two references it is given in title case!
50. Line 379. Give full details of page numbers. If it is a open access journal without volume or page numbers, give DOI.
51. Line 381. 'The plant cell'???
52. Lines 383-384. Title of paper is incomplete. 'genes in trans.....'??. Page numbers missing.
53. Line 395. Full name of journal is given. USA is missing. End page number is missing.
54. Line 397. Page numbers missing.
55. Line 407. Check Peer J format. Delete symbols like %, @ etc.
56. Line 413. Change to 'MicroRNA'
57. Line 416. Change 'U S A' to 'USA'
58. Line 421. Page numbers missing.
59. Fig. 2 Title: Small RNA sequencing quality analysis... carrys no meaning. Introduce hyphen "-" between 'sequencing' and 'quality'.
60. Fig. 3. Remove "Change of". Remove period at the end of figure title.
61. Fig. 3. "When alignmenting'..??? Change to 'when aligning with' and change it in all the three places in the figure legend.
62. Fig. 5. Remove 'difference'. Make the words ' analysis of' as first two words of the figure title. Do appropriate modifications in the legend.

Reviewer 2 ·

Basic reporting

In general MS is clear and written in standard language. The background information on introduction, clearly indicated the importance of the study, well strengthen by updated relevant references. However, if authors can improve the significance of the study by focusing their results to emphasize the validity of their results would be more attractive to the readers. Specifically line number 76-83.

Experimental design

The study is original and novel. Methods describe with sufficient details. Standard protocols were used. Thorough experiments performed in a scientifically standard level. Research problem well defined, applicable & expressive. It address the importance of this research to fill the identified knowledge gap.

Validity of the findings

No indication of field experimental layout and replications. But, have solid and statistically stable data with novel findings. However, the meaning of the explained results particularly in lines of 139-145 ambiguous and need improvements.

Additional comments

I appreciate the authors for their solid data set, assembled over F1 to F2 generations of detailed fieldwork. In general, the manuscript is clearly written in proficient, unambiguous language. However, there are certain poorly written places as previously indicated, which should be improved .

·

Basic reporting

The submitted article is clear and understandable to the reader. The language is good. Literature review is good and the references are appropriate. This manuscript contains suitable Figures, tables, data and relevant results.

Experimental design

The submitted work comes under the scope of the journal. The research question is well defined and the research investigations are performed with good standard. The methodology used in this research and the experimental design are appropriate.

Validity of the findings

This article contains original, novel and interesting results. The data provided in the experiments are robust and statistically sound.

Additional comments

The submitted manuscript contains some original and useful information. The submitted work has studied the sRNA changes and regulatory patterns of allopolyploids with reference to the flowering trait. The experimental design is good and the results are reliable. The manuscript is clearly written. Hence this work is suitable for Publication with following minor corrections.

Abstract: Abstract is comprehensive and informative.
Introduction : Introduction is clear enough to introduce the reader. The aim and objective of the study are clear. However the authors failed to write why they select the sRNA changes and regulatory patterns of flowering trait and it's importance.

Materials and Methods : The research methodology are clearly written and appropriate.

Results : Results are clear with supporting evidences. The presentations of Figures are with good quality. However the Legends for Fig.3c and 3d are looks same. Please check the legends.
Line No: 218 Write the full form for 'FT"

Discussion : Discussions are sound and enough.

Minor concern :
Spelling mistakes
Line No: 156 the word Known is misspelled.
Line No : 188 the word indicate is misspelled

---

## Round 0.2 · accepted · Accept

As the required revision been carried out, the manuscript can be accepted for publications

·

Basic reporting

No comment.

Experimental design

No comment.

Validity of the findings

No comment.

Additional comments

The authors have revised the manuscript and it reads acceptable.